# Rapid and Portable Detection of Hg and Cd in Grain Samples Based on Novel Catalytic Pyrolysis Composite Trap Coupled with Miniature Atomic Absorption Spectrometry

**DOI:** 10.3390/foods12091778

**Published:** 2023-04-25

**Authors:** Tengpeng Liu, Jixin Liu, Xuefei Mao, Xiaoming Jiang, Yabo Zhao, Yongzhong Qian

**Affiliations:** 1Institute of Quality Standard and Testing Technology for Agro-Products, Chinese Academy of Agricultural Sciences, and Key Laboratory of Agro-Food Safety and Quality, Ministry of Agriculture and Rural Affairs, Beijing 100081, China; ltp101028@163.com (T.L.);; 2Beijing Ability Technology Co., Ltd., Beijing 100081, China; 3Analytical & Testing Center, Sichuan University, Chengdu 610064, China

**Keywords:** mercury, cadmium, grain, portable determination, solid sampling

## Abstract

As toxic metals, Hg and Cd are a concern for food safety and human health; their rapid and portable analysis is still a challenge. A portable and rapid Hg–Cd analyzer constructed from a metal–ceramic heater (MCH)-based electrothermal vaporizer (ETV), an on-line catalytic pyrolysis furnace (CPF), a composite *Pt/Ni* trap, and a homemade miniature atomic absorption spectrometer (AAS) was proposed for grain analysis in this work. To enhance sensitivity, a new folded light path was designed for simultaneous Hg and Cd analysis using charge coupled device (CCD) in AAS. To eliminate the grain matrix interference, a catalytic pyrolysis furnace with aluminum oxide fillers was utilized to couple with a composite *Pt/Ni* trap. The method limits of detection (LODs) were 1.1 μg/kg and 0.3 μg/kg for Hg and Cd using a 20 mg grain sample, fulfilling the real sample analysis to monitor the grain contamination quickly; linearity *R^2^* > 0.995 was reached only using standard solution calibration, indicating the sample was free of grain matrix interference. The favorable analytical accuracy and precision were validated by analyzing real and certified reference material (CRM) grains with recoveries of 97–103% and 96–111% for Hg and Cd, respectively. The total analysis time was less than 5 min without sample digestion or use of any chemicals, and the instrumental size and power consumption were <14 kg and 270 W, respectively. Compared with other rapid methods, this newly designed Hg–Cd analyzer is proven to be simple, portable, and robust and is, thus, suitable to quickly monitor Hg and Cd contamination in the field to protect grain and food safety.

## 1. Introduction

In recent years, heavy metals have become the main contaminants in the environment and agri-foods due to the rapid industrialization, urbanization, and intensification of agriculture [1,2]. Heavy metals threaten food safety and can lead to human health risks such as neurotoxicity, teratogenicity, and cancer. Among them, cadmium (Cd) is classified as Group 1 carcinogen that damages human bone, kidney, and ocular tissues [3,4], and mercury (Hg) is a dangerous pollutant that also causes damage to the brain, cardiovascular, and nervous systems [5,6]. Owing to their persistent, nonbiodegradable, and accumulative properties, food intake has been recognized as the major pathway for human exposure. Cd’s maximum allowable concentration (MAC) in grain has been established at 0.10 mg/kg by the European Commission (EC) and at 0.2 mg/kg for rice by China, and the MAC of Hg has been set at 0.02 mg/kg by China [7,8], respectively. Hence, to protect grain safety and human health, fast and precise monitoring of Hg and Cd in crop planting, grain processing, and food marketing is of great significance. 

Conventional atomic spectrometric techniques, including inductively coupled plasma mass spectrometry/optical emission spectrometry (ICP-MS/OES) [9,10], atomic absorption spectrometry (AAS) [11], and atomic fluorescence spectrometry (AFS) [12] are well known for their excellent detection limits (LODs), sensitivity, and stability in elemental analysis for agri-foods. However, these approaches typically require complex sample digestion, which is time-consuming, costly, uses toxic reagents, and suffers possible contamination, resulting in failure in rapid and field detection. In addition, X-ray fluorescence spectrometry (XRF) and laser-induced breakdown spectroscopy (LIBS) are also suitable for portable, rapid, and multi-elemental analysis. However, they are limited due to the inadequate LODs at the mg/kg level and the difficulty in accurately detecting trace Hg and Cd in grains [13,14]. Therefore, it is worthwhile to explore a portable and miniaturized analyzer for the rapid and portable determination of Hg and Cd in solid grain samples without any digestion processes. 

Furthermore, direct sampling (DS) is also a good alternative for the direct analysis of solid grain samples. As an excellent DS method, electrothermal vaporization (ETV) has been used for the hyphenation of various atomic spectrometers or inorganic mass spectrometers due to its high sample introduction efficiency and exceptional versatility [15,16]. The earliest ETV devices were derived from graphite furnaces, but they are unsuited for miniature and portable instrument due to the need for a large power supply and water-cooling system [17]. Thus, to miniaturize the ETV’s size, high melting point metals have been fabricated as strips, coils, rods, or furnaces to vaporize target elements [18,19]. Among them, tungsten coil (W-coil) has been commonly utilized due to its small size, easy operation, and low power consumption [20]. However, its spiral structure only accommodates liquid or slurry mediums, therefore, cannot directly undertake solid samples. Lv et al. first enabled the direct introduction of Hg into a solid soil sample using an in-built metal–ceramic heater (MCH) that only consumes 30 W for up to an 80 mg sample size [21]. Thus, this vaporizer may be ideal for rapid detection and field monitoring due to its low power and small size. Subsequently, the co-introduction of Hg and Cd via 100 W heating based on AFS detection was achieved for soil analysis; however, the AFS detector design complicates the apparatus and enlarges the instrument size, resulting in the impossibility of miniaturization and portability for the Hg–Cd analyzer [22]. 

On the other hand, the direct sampling analysis of grain samples is still a challenge in view of the presence of carbohydrates, lipids, proteins, and other matrices. The matrix interferents and microparticles, accompanied with vaporized analytes, might influence the vaporization, transportation, and atomization of Hg and Cd, leading to analytical inaccuracy and imprecision. Gas phase enrichment (GPE) is an effective and available approach to eliminate matrix interference via preconcentrating and separating targeted elements from the gaseous interferents. Herein, trapping materials such as gold [23], quartz [24,25], TC trap (TCT) [26,27], and foamed nickel trap (NT) [28] have been frequently utilized to couple with atomic spectrometers. For instance, as one of the most successful GPE tools, the gold amalgamator is widely used in various direct sampling Hg-analyzers, which have also been adopted as the standard method in the USA (EPA 7473) and China (HJ 923–2017). To detect Hg and Cd during one sampling cycle, a gold amalgamator and a TCT were assembled to trap Hg and Cd sequentially and to separately release them [23]. However, the separate traps result in a low-level integration for the Hg–Cd analyzer. To further miniaturize the co-trapping device, a *Pt/Ni* composite trap was fabricated for simultaneous on-line trapping of Hg and Cd and specially designed for solid soil analysis; however, food matrices containing organic substances are more complicated than soil and interfere with the signals of Hg and Cd [22]. In fact, the original *Pt/Ni* composite trap without the integrated catalytic pyrolysis used in this study does not work well for the precise analysis of grain samples.

To fulfill the simultaneous detection of Hg and Cd, atomic spectrometers must be improved in terms of the light source, light path, and detector [29,30,31]. Cold vapor (CV) AAS is well known for its stability and simplicity and has frequently been employed in commercial Hg analyzers [32,33]. In theory, CV-AAS is also suitable for the detection of Hg and Cd due to the release of atoms from the *Pt/Ni* composite trap. However, the original CV-AAS always incorporates a photomultiplier (PMT) as the signal sensor, which is incapable of simultaneous multi-element analysis [21,34,35,36]. A fiber optical spectrometer (FOS) could therefore be a favorable alternative, although it suffers from inferior sensitivity compared to PMT. To remedy the lack of sensitivity, the light path might be lengthened [37,38]. Until now, the application of the simultaneous determination of Hg and Cd on the basis of integrated AAS has not been reported.

In this study, for the rapid and simultaneous detection of Hg and Cd in grains, a portable Hg–Cd analyzer was constructed using an MCH-ETV, a novel catalytic pyrolysis *Pt/Ni* trap, and a miniaturized AAS. Some key parameters, such as the effect of MCH vaporization, on-line ashing of catalytic pyrolysis, and enrichment of the *Pt/Ni* trap and miniaturized AAS, were evaluated; additionally, the analytical figures of merit and real grain sample analysis were obtained. The whole analysis time was less than 5 min, and the instrumental size and power consumption were <14 kg and 270 W. Thus, this proposed method of simultaneous Hg and Cd detection demonstrated rapidness, sensitivity, simplicity, and portability with promising applications in the future.

## 2. Experimental

### 2.1. Instrumentation

As shown in Figure 1a, the portable Hg–Cd analyzer primarily consists of a sample introduction module and a detection module. The sample introduction module mainly includes an MCH coupled with a catalytic pyrolysis furnace (CPF), in which MCH acts as a sampling boat as well as a vaporizer similar to that described in our previous work [21]. To meet the food matrices, a CPF filled with the Al_2_O_3_ was coupled to the MCH-ETV to eliminate organic interference; otherwise, to eliminate inorganic interference, a composite *Pt/Ni* trap was utilized as a GPE unit for the simultaneous trapping of Hg and Cd to separate other interferents. Herein, the composite *Pt/Ni* trap was modified for *Pt/Ni* wafers according to the grain matrix effect. In addition, air and Ar/H_2_ (v:v = 9:1) as the working gases were also provided as required. 

As shown in Figure 1b, the detection module was connected to the *Pt/Ni* trap, in which a miniaturized AAS was newly laboratory-made and equipped with 253.7 nm and 228.8 nm hollow cathode lamps (HCL, Beijing Shuguangming Electronic Lighting Instrument Co., Ltd., Beijing, China) as Hg and Cd light sources, respectively, and a charge coupled device (CCD) spectrometer (Maya 2000 Pro, 185–400 nm, Ocean Optics, Orlando, FL, USA) as the detector. To improve the sensitivity, a folded optical path (length 130 mm × 2) was utilized to prolong the light path, consisting of two parallel quartz tubes with a quartz inlet and outlet in each of the two quartz tubes for the transmission of the analytes. In addition, an optical system was fixed between the HCL and the CCD spectrometer to allow the characteristic lights of the Hg and Cd to pass through, which mainly consists of a mirror (diameter 30 mm), a dichroic mirror (diameter 30 mm), a folded optical path, two convex lenses (diameter 11 mm), and two concave mirrors (length 13 mm). The mirror was utilized to reflect the characteristic light of the Hg lamp, and the dichroic mirror was utilized to reflect the characteristic light of the Cd lamp while transmitting the characteristic light of the Hg lamp at the same time. The model instrumental picture of the portable Hg–Cd analyzer and the miniature AAS are shown in Appendix A.

To validate the accuracy of the proposed Hg–Cd analyzer, grain samples were introduced to detect Hg and Cd with the Hg analyzer (Model 5E-HGT2321, Changsha Kaiyuan Hongsheng Technology Co., Ltd., Changsha, China) and microwave digestion ICP-MS (Agilent 7900, Agilent Technologies, Santa Clara, CA, USA), respectively. The detailed instrumental conditions of the Hg analyzer, ICP-MS, and microwave digestion system are summarized in Appendix A.

### 2.2. Reagents, Materials, and Samples

All chemicals used were of analytical grade or better and purchased from Shanghai Macklin Biochemical Co., Ltd. (Shanghai, China). Standard stock solutions 1000 mg/L Hg, Cd, As, Pb, Zn, Cr, Mn, Ni, Fe, Cu, Mg, Ca, and Al were obtained from the National Research Center for Certified Reference Materials (NRCCRM) (Beijing, China). Working standard solutions were stepwise diluted in purified water prepared using a Milli-Q integral water purification system (Millipore, Billerica, MA, USA).

The certified reference materials (CRMs) of grains, including rice CRMs, GBW(E)100351 (certified Hg = 3.6 ± 0.9 µg/kg, certified Cd = 420 ± 20 µg/kg), GBW(E)100360 (certified Hg = 3 ± 1 µg/kg, certified Cd = 220 ± 20 µg/kg), wheat CRM, GSB-24 (GBW10046, certified Hg = 2.2 µg/kg, certified Cd = 18 ± 2 µg/kg), and corn CRM, GSB-3 (GBW10012, certified Hg = 1.6 µg/kg, certified Cd = 4.1 ± 1.6 µg/kg) were obtained from the NRCCRM. The real grain samples were obtained from the local markets (Beijing, China) and ground, sieved through a 60-mesh sieve, then stored at room temperature for analysis.

### 2.3. Analytical Procedures

The detection procedure for the portable Hg–Cd analyzer was as follows. (1) The catalytic pyrolysis furnace was first preheated (700 °C) for 10 min. (2) A total of 5–20 mg of grain powder was placed into the MCH and inserted into the ETV chamber. (3) The sample was heated at 30 W (~450 °C) for 40 s in air, during which the dehydration and ashing, Hg vaporization, transportation, and trapping were performed. (4) The sample was then ashed at 50 W (~550 °C) for 100 s to remove residual organic substances in the sample matrix and to vaporize residual Hg under an air atmosphere. (5) Subsequently, the carrier gas was switched to Ar/H_2_, and the MCH was heated at 100 W (~950 °C) for 40 s to vaporize Cd; the Cd was transported to the *Pt/Ni* trap and trapped on the *Pt/Ni* materials. (6) Then, the *Pt/Ni* trap was heated at 140 W (~1000 °C) for 25 s under Ar/H_2_ to release Hg and Cd simultaneously. (7) Finally, the released Hg and Cd were transported into the AAS for measurement by the carrier gas. The program sequence of the portable Hg–Cd analyzer is shown in Table 1.

## 3. Results and Discussion

### 3.1. Design and Effect of Miniaturized AAS

Based on the CCD detector, multi-elemental analysis can be achieved by AAS, e.g., commercialized high-resolution continuous source (HR-CS) AAS ContrAA 800 manufactured by Jena, Germany. However, the above AAS instruments are bulky, high-powered, and costly; additionally, the maximum sampling size is always limited to 1 mg due to serious matrix interference. Thus, first, the size of the AAS instrument must be minimized as much as possible to fulfill the field deployment. In this work, due to the elimination of matrix interference by the catalytic pyrolysis *Pt/Ni* trap, echelle grating, background correction, e.g., Zeeman effect, and deuterium lamp used in HR-CS-AAS can be canceled. Therefore, only Hg and Cd HCL, a folded light path, and CCD spectrometer were retained, as shown in Figure 1b and Appendix A, and the instrumental size was thereby minimized. This homemade AAS apparatus occupies 40 cm × 35 cm × 15 cm outside size and weighs ~9 kg, and the power consumption can be controlled within 30 W. 

To verify this newly designed AAS, the background spectrum was acquired, as shown in Appendix A, in which the analytical lines at 253.998 nm and 229.09 nm were used for Hg and Cd, respectively. To obtain higher analytical sensitivity, a folded optical path was utilized to prolong the optical path length. Therefore, in Figure 2, the Hg and Cd signal intensities using the folded optical path were significantly enhanced by 35% and 57%, respectively, compared to the signal optical path. In addition to the sensitivity, the stability of this newly designed AAS was demonstrated for Hg and Cd signal acquisition in the following experiments. As a result, this proposed miniature AAS apparatus is available for the portable Hg–Cd analyzer. 

### 3.2. Effect of ETV and Catalytic Unit for Grain Samples

In the previous work [22], the MCH-*Pt/Ni* trap system worked well for precise soil analysis. However, as shown in Figure 3, when introducing a 10 mg grain sample, the peak heights of Hg and Cd without CPF were obviously lower than those with the catalytic pyrolysis furnace, presumably due to matrix interference from the grain sample, especially the presence of organic substances. Herein, with sample ashing and vaporization, gaseous organics or solid micro-particles from grain matrices entered the *Pt/Ni* trap and covered the material surface, thereby partly failing to form an alloy of *Pt/Ni* trap with Hg and Cd, and leading to signal intensity decline. Furthermore, the absence of CPF resulted in split and irregular peaks, which were also detrimental to precise and accurate detection for Hg and Cd. To fulfill the analysis of the food matrix, the catalytic pyrolysis furnace has been assembled into the MCH-ETV unit to decompose and absorb the gaseous organics or solid micro-particles so that the *Pt/Ni* trap obtains “clean” gaseous analyte during the GPE process. Furthermore, to facilitate the transport of Cd atoms during Cd vaporization, the temperature of the CPF was also investigated. In Appendix A, when below 700 °C, the Cd signal intensity increased with the temperature, and the signal reached a platform with favorable relative standard deviations (RSDs) from 700–800 °C, which may be due to some Cd atoms being retained in the CPF during the low temperature. Thus, the temperature of the CPF was maintained at 700 °C.

To investigate the effect of MCH-ETV on Hg and Cd vaporization, 500 μg/L mixed Hg and Cd solution (10 μL) was introduced into the MCH-CPF-*Pt/Ni*-AAS. As shown in Figure 4a, as the power increases, the Hg signals rapidly increase and reach a plateau from 30 W; meanwhile, the Cd signals begin to rise at 50 W and reach a plateau from 100 W. Further, to eliminate the matrix interference, the CPF was maintained at 700 °C to keep the transport of Hg and Cd analytes [21,23]. The vaporization trends of Hg and Cd in the grain sample using MCH-ETV are shown in Appendix A. Heating at 30 W for 40 s fulfilled the Hg vaporization, in which sample dehydration and ashing processes occurred; then, 50 W heating for 100 s continued ashing to remove organic substances from the residual sample completely; finally, 100 W for 40 s enabled 100% vaporization of Cd for measurement. As shown in Figure 4b, ashing time ranging from 60 s to 110 s was investigated. The Hg and Cd signals substantially increased with the increase of the ashing time and plateaued at 90 s, which implied complete Hg and Cd vaporization, according to previous studies [15,16,21]. These results demonstrate that the ETV programs in Table 1 are adequate for the following experiments.

### 3.3. Hg and Cd Trapping and Releasing by Pt/Ni Trap 

The preconcentration of Hg and Cd is also essential to enhance sensitivity and eliminate interference for direct sampling. Our previous study [22] has demonstrated the co-trapping and releasing of Hg and Cd by the foamed platinum–nickel wafers. Considering the redesigned ETV system and grain sample matrix, the trapping, release, and working gas conditions need to be reinvestigated in this work. For trapping, a 20 ng Hg and Cd mixed solution was introduced into the MCH-CPF-*Pt/Ni*-AAS, as shown in Figure 5a, the number of wafers in the *Pt/Ni* trap was reinvestigated. The breakthrough signals decreased with the number of wafers and reached 0 at 8 *Pt/Ni* wafers, indicating the complete co-trapping of Hg and Cd. Thus, at least 8 *Pt/Ni* wafers were applicable to assemble for the *Pt/Ni* trap in the following experiment. Further, as shown in Appendix A, the maximum Hg and Cd capture capacities of these 8 *Pt/Ni* wafers were also demonstrated to be 30 ng, implying 3 mg/kg for a 10 mg sample size and fulfilling real grain analysis. 

The heating power and duration of the *Pt/Ni* trap were optimized for releasing. As shown in Figure 5b, with the increase of power, the released Hg and Cd signals rose rapidly and reached a platform from 140 W to 180 W. Moreover, the Hg and Cd intensities also reached a platform with more than 15 s and 25 s release time (in Appendix A), respectively. Consequently, heating 140 W for 25 s was utilized for Hg and Cd release.

### 3.4. Working Gas

The working gas is also essential for analytes vaporization and transport, in which oxygen is mainly responsible for sample ashing and Hg transportation, and Ar/H_2_ is for Cd vaporization and transportation under a reducing atmosphere. Herein, due to the effect of H radicals, a reducing atmosphere is beneficial for the decline of Cd vaporization temperature [39]. Considering the presence of complicated organic substances, air is more important for grain analysis compared with soil samples. In Appendix A, the air flow rates are also investigated. Both Hg and Cd signals tend to be stable from 300 to 400 mL/min, so the air was set at 300 mL/min for Hg trapping and sample ashing.

For Cd trapping, as shown in Appendix A, the highest signal appeared at 600 mL/min. Thus, the flow rate of Ar/H_2_ was selected at 600 mL/min for Cd trapping. Furthermore, Appendix A demonstrates the Hg and Cd release under Ar/H_2_; the absorbances of Hg and Cd were the highest at 800 mL/min and then decreased slightly due to the dilution of superfluous carrier gas. Therefore, 800 mL/min Ar/H_2_ was chosen for Hg and Cd releasing in the remaining experiments.

### 3.5. Interference Study

The possible interferences from inorganic elements were evaluated by spiking a 10 mg rice sample (R-03) with coexisting ions (As, Pb, Zn, Cr, Mn, Ni, Fe, Cu, Mg, Ca, and Al) using the proposed method, respectively. As shown in Appendix A, the measured recoveries for Hg and Cd were all within 89–115%, indicating that there was no significant interference from the inorganic elements listed below under these levels. Furthermore, to verify the anti-interference ability of this method, the signal peaks of the 10 mg rice sample (R-03) were compared with the same Hg and Cd concentrations of the standard solution. As shown in Figure 6, both the Hg and Cd peaks of the rice sample have a slight delay compared to the standard solution. However, the peak heights between the rice sample and standard solution were approximately consistent (0.96:1 for Hg and 0.99:1 for Cd), proving that there were no interferences in Hg and Cd analysis. Compared with peak area, peak height is more competent for precise analysis of grain samples using a standard solution calibration strategy. Thus, the proposed MCH-CPF-*Pt/Ni*-AAS method has favorable anti-interference, which is appropriate for detecting Hg and Cd by direct solid sampling.

### 3.6. Analytical Performance of the Method and Real Sample Analysis

In conclusion, Figure 6 shows that there is no significant difference in signal peak height between the solid sample and the standard solution; the demonstrated matrix interference of grain samples is able to be completely eliminated using the MCH-CPF-*Pt/Ni*-AAS method. The standard calibration strategy was, therefore, performed to directly analyze the Hg and Cd, which facilitates the instrument’s operation in the real world for solid sampling. Moreover, the Hg and Cd calibration curves were evaluated by measuring the mixed solutions ranging from 0.05 μg/L to 1 μg/L (10 μL). The obtained regression equations were Hg: Y = 0.0003X + 0.0025 with a linear coefficient (*R^2^*) of 0.9956; Cd: Y = 0.0007X + 0.0158 with *R^2^* of 0.9976, respectively. The LODs of Hg and Cd were calculated by taking 3 times the standard deviation of 11 repeated measurements divided by the slope, which were found to be 21 pg and 6 pg, namely 1.1 μg/kg and 0.3 μg/kg for a 20 mg sample size, respectively; moreover, the limit of quantifications (LOQs) were 3.6 μg/kg and 1.0 μg/kg, respectively. The RSDs of repeated 5 measurements are less than 10% and 20% for Hg and Cd in standard solutions (10 μL Hg and Cd mix solution, 500 μg/L) and grain samples (10 mg R-3 rice sample, Hg: 44 μg/kg, Cd: 870 μg/kg), respectively.

Sample amounts were also evaluated, ranging from 2 mg to 20 mg (R-03, n = 5), and the results are shown in Appendix A. Less than 5 mg resulted in poor accuracy and precision due to insufficient sample homogeneity of Hg and Cd; the results in 5–20 mg were within the certified values of CRMs. Thus, for real grain analysis, taking into account sample representativeness, 5 mg or more can be used after good homogenization (60 mesh sieving or better).

The accuracy of the proposed MCH-CPF-*Pt/Ni*-AAS method was confirmed by the analysis of different grain samples and compared with the standard methods. In Table 2, the results of CRMs and real grain samples were consistent with certified values and determined by the Hg analyzer or ICP-MS (*p* > 0.05, Student’s *t*-test), and the Hg and Cd recoveries of real grains ranged from 97% to 103% and 96% to 111%, respectively, indicating a reliable method accuracy and feasibility.

In comparison with previously reported Hg or Cd detection methods, as listed in Table 3 [16,23,39,40,41,42,43,44,45,46], the Hg and Cd LODs of this work are slightly higher or comparable. However, this proposed instrumentation fulfills the simultaneous detection of Hg and Cd and consumes only <270 W and 14 kg due to the use of a composite trap and miniature AAS module. In the previous study [23], only one report fabricated a Hg–Cd analyzer using AFS; however, its size and power consumption are more than 60 kg and 1500 W, respectively. In contrast, this proposed Hg–Cd analyzer is obviously superior to other analyzers and is, therefore, appropriate for rapid Hg and Cd analysis of grain samples in the field.

## 4. Conclusions

In this study, a rapid and simple method for the simultaneous direct analysis of Hg and Cd in grain samples was investigated based on a laboratory-made portable Hg–Cd analyzer combined with miniaturized AAS. An on-line catalytic pyrolysis furnace coupled with the composite *Pt/Ni* trap was utilized to eliminate grain matrix interference. For various grain samples, this proposed method was demonstrated to be free from matrix interference, allowing the standard solution calibration to be directly employed independent of standard addition or matrix matching methods. This portable Hg–Cd analyzer consumes only <270 W and 14 kg due to the use of a composite trap and miniature AAS module. Furthermore, this method only consumes < 5 min without a complicated sample digestion process and use of any chemicals; the LODs of Hg and Cd fulfill the real sample analysis to quickly monitor Hg and Cd contamination in grain. In a word, this newly designed Hg–Cd analyzer has been proven to be rapid, sensitive, portable, simple, and robust, and thus suitable for fast and field detection of Hg and Cd detection in grain to protect food safety.

## Figures and Tables

**Figure 1 foods-12-01778-f001:**
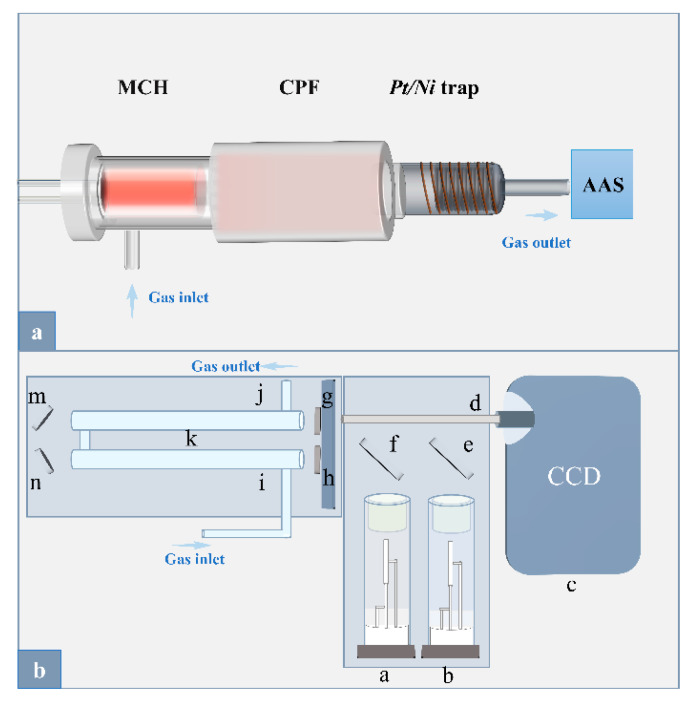
System arrangement of the portable direct sampling Hg–Cd analyzer. (Panel **a**): the schematic diagram of the solid sampling module. (Panel **b**): the schematic diagram of the detection module. a: Cd hollow cathode lamp (operating current: 5 mA), b: Hg hollow cathode lamp (operating current: 3 mA), c: CCD spectrometer (integration time: 200 ms), d: fiber optic, e: mirror, f: dichroic mirror, h/g: convex lenses, k: folded optical path, m/n: concave mirrors, i: gas inlet; j: exhaust outlet.

**Figure 2 foods-12-01778-f002:**
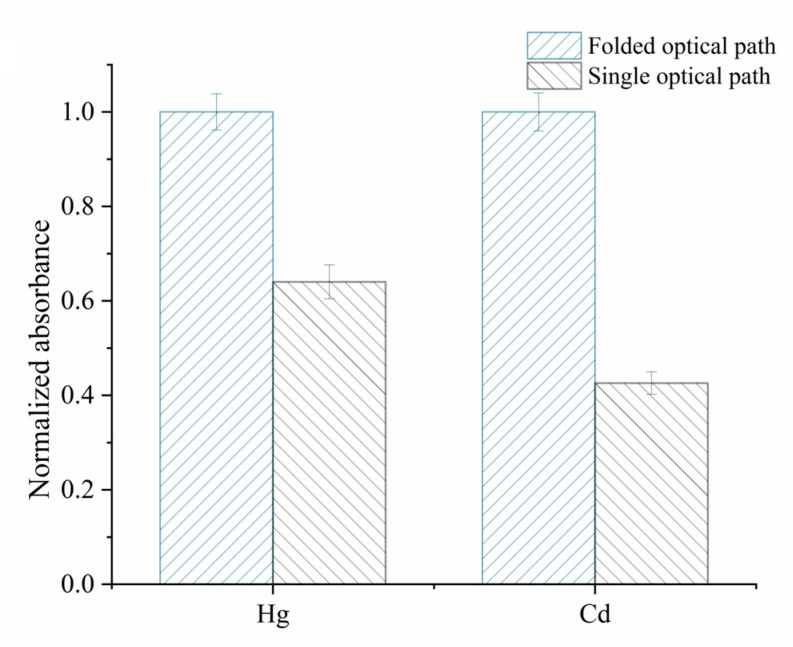
Effect of folded optical path on AAS signals of Hg and Cd. The Hg and Cd signals were obtained under the different optical path lengths, of which the absorbances are normalized with the signal at the folded optical path and set as 1. A 10 μL mixed Hg and Cd mixed solution (500 μg/L) was introduced into the MCH-CPF-*Pt/Ni*-AAS for measurement under different optical path lengths.

**Figure 3 foods-12-01778-f003:**
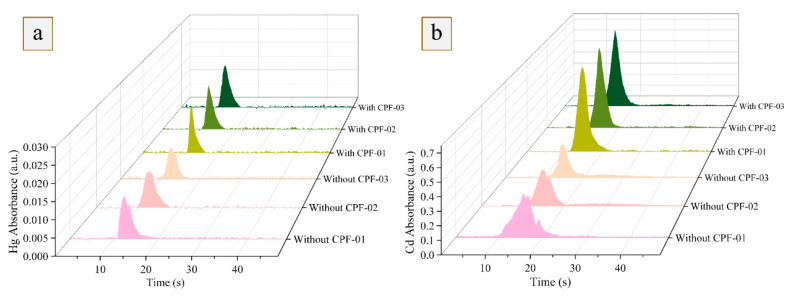
Comparison of the releasing peaks of Hg and Cd in grain samples by the MCH-*Pt/Ni*-AAS with and without catalytic pyrolysis furnace (n = 3). Herein, a 10 mg rice sample (R-03) was introduced for measurement. (Panel **a**): Hg; (Panel **b**): Cd.

**Figure 4 foods-12-01778-f004:**
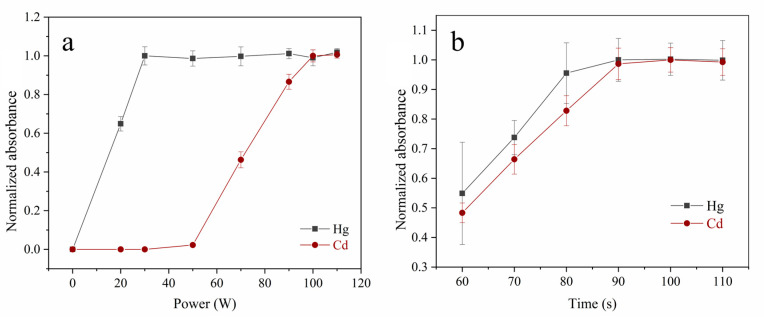
The effect of heating power and time of the MCH-ETV on Hg and Cd detection (n = 3). (Panel **a**): The Hg and Cd vaporization power by MCH. A 10 μL Hg and Cd mixed solution (500 μg/L) was introduced into the MCH-CPF-*Pt/Ni*-AAS. The absorbances are normalized with the signals at 30 W and 100 W set as 1 for Hg and Cd, respectively. (Panel **b**): The Hg and Cd absorbance under the different ashing time. Herein, a 10 mg R-03 sample was introduced into the MCH-ETV-*Pt/Ni*-AAS. The absorbances are normalized with the signals at 90 s and 100 s set as 1 for Hg and Cd, respectively.

**Figure 5 foods-12-01778-f005:**
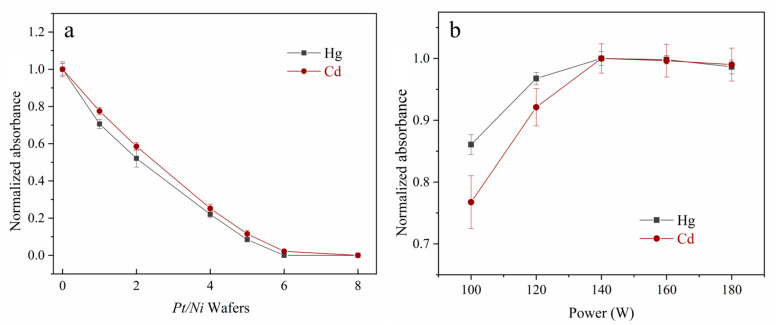
Trapping and release effects of the *Pt/Ni* trap (n = 3). (Panel **a**): The amounts of *Pt/Ni* wafers used to trap Hg and Cd. The breakthrough intensities are both normalized with signals at 0 set as 1, for Hg and Cd, respectively. (Panel **b**): The heating power for Hg and Cd releasing. The absorbances are both normalized with the signal at 140 W set as 1 for Hg and Cd, respectively.

**Figure 6 foods-12-01778-f006:**
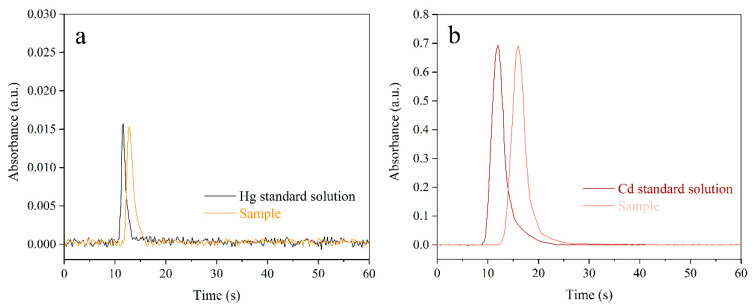
Comparison of the Hg and Cd signal peaks using the same concentration of the standard solution and real rice sample. (Panel **a**): The Hg signal peaks of the standard solution (44 μg/kg) and rice sample (R-03) with 44 μg/kg Hg. (Panel **b**): The Cd signal peaks of the standard solution (870 μg/kg) and rice sample (R-03) with 870 μg/kg Cd.

**Table 1 foods-12-01778-t001:** Programs of the portable Hg–Cd analyzer.

Program	Power/W	Time/s	Gas	Signal Acquisition
Dehydration/Hg Vaporization	30	40	Air	No
Further ashing	50	100	Air	No
Gas switching	0	15	Ar/H_2_	No
Cd vaporization	100	40	Ar/H_2_	No
Release	140	25	Ar/H_2_	Yes
Clean	150	40	Ar/H_2_	No

**Table 2 foods-12-01778-t002:** The presence and recoveries of Hg and Cd in real grain samples by the proposed method (mean ± standard deviation, n = 3).

Samples	Certified or Standard Method *^a^* (µg/kg)		This Method (µg/kg)		Added (μg/kg)		Measured Values (µg/kg)		Average Recoveries (%) *^b^*	
	Hg	Cd	Hg	Cd	Hg	Cd	Hg	Cd	Hg	Cd
Rice (GBW(E)100351)	3.6 ± 0.9	420 ± 20	3.8 ± 0.8	419 ± 17	/	/	/	/	/	/
Rice (GBW(E)100360)	3 ± 1	220 ± 20	3.6 ± 0.7	226 ± 15	/	/	/	/	/	/
Wheat (GSB-24)	2.4 ± 0.3	18 ± 2	*NQ*	18 ± 1	10	10	13 ± 1	28 ± 2	-	98
Corn (GSB-3)	1.7 ± 0.2	4.1 ± 1.6	*NQ*	4.1 ± 0.6	10	10	12 ± 1	15 ± 1	-	106
Rice (R-01)	2.4 ± 0.4	56 ± 3	*NQ*	54 ± 3	10	100	12 ± 1	160 ± 7	-	106
Rice (R-02)	22 ± 1	433 ± 25	22 ± 2	444 ± 11	50	500	71 ± 4	968 ± 60	97	105
Rice (R-03)	44 ± 1	870 ± 26	44 ± 2	880 ± 28	50	500	94 ± 6	1356 ± 65	101	96
Wheat (W-01)	7.4 ± 0.5	106 ± 3	7.1 ± 0.8	104 ± 3	10	100	18 ± 1	215 ± 6	102	111
Corn (C-01)	12 ± 1	226 ± 4	12 ± 1	224 ± 5	50	500	64 ± 4	748 ± 20	103	105

*NQ* means below the method LOQ. *^a^* Measured Hg value using the commercial Hg analyzer; measured Cd value using the microwave digestion ICPMS method. *^b^* Recovery is the ratio of the spiked Hg and Cd levels measured with the proposed method after spiking to the added values of Hg and Cd.

**Table 3 foods-12-01778-t003:** Comparison of analytical performances of this proposed method with others reported.

Method	Elements	Sampling Mode	Method LOD (μg/kg)	RSD (%)	Sample	Sample Size (mg)	Instrumental Weight and Power	Instrument Model and Details	Analysis Time (min)	Refs.
CPE-CV-AAS	Hg	Liquid	0.117	<5	Broiler chicken	1.5 mL	200 kg; 5 kW	Aanalyst700; Desktop	/	[40]
HR CS GFAAS	Hg	Liquid	2.3	15	Blood, urine	/	200 kg; 2.1 kW	ContrAA700; Desktop	<2	[41]
ETV-DBD-GT-AFS	Hg	Solid	0.5	10	Aquatic food	2–12	60 kg; 1.3 kW	Modified from DCMA200; Desktop	5	[42]
Direct mercury analyzer	Hg	Solid	LOQ: 0.6	<5.8	Food	300	65 kg; 1.8 kW	AMA 254; Desktop	<6	[43]
Direct mercury analyzer	Hg	Solid	1 ng	/	Fish	50	56 kg; 2 kw	DMA-80; Desktop	<5	[44]
CV-AAS	Cd	Liquid	2.1 ng/L	2.5	Water	2.8 mL	200 kg; 5 kW	Aanalyst700 Desktop	/	[45]
GF AAS	Cd	Slurry	0.09	20	Chocolate	100	200 kg; 2 kW	ContrAA800 Desktop	<2	[46]
ETV-AAS	Cd	Solid	0.015	10	Grain	200	40 kg; 2 kW	Homemade Desktop	3	[39]
ETV-TC-APGD-AES	Cd	Liquid	11.9	5.8	Rice	5 μL	4.5 kg; 37 W	Homemade; Portable	3	[16]
Hg–Cd analyzer (Tungsten and gold coil traps) AFS	Hg, Cd	Solid	Hg: 0.07; Cd: 0.05	15	Food	10	60 kg; 1.5 kW	DCMA200; Desktop	3	[23]
MCH-CPF-*Pt/Ni*-AAS	Hg, Cd	Solid	Hg: 1.1; Cd: 0.3	20	Grain	15	14 kg; 270 W	Homemade; Portable	5	This work

## Data Availability

All data were shown in the article and Appendix A.

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
