# Peer review of "Rapid and Portable Detection of Hg and Cd in Grain Samples Based on Novel Catalytic Pyrolysis Composite Trap Coupled with Miniature Atomic Absorption Spectrometry"

_foods, 2023, doi:10.3390/foods12091778_

Round 1

Reviewer 1 Report

The presented work represents a significant technological advancement for the direct determination of Hg and Cd in grain samples. Additionally, it was carried out very rigorously, and all relevant figures of merit were determined, including the trueness, precision, calibration parameters such as LOD and LOQ, as well as robustness studies.

The obtained results, both in terms of technological level and analytical performance, are very good. The originality of incorporating a catalytic system to eliminate organic interferences, including 2 HCL lamps, one for each element, and a CCD detector for simultaneous detection, as well as increasing the optical path to improve sensitivity, are technological achievements that represent a significant advance, and the comparison with previous works is well illustrated.

Most works of this type focus exclusively on the determination of Hg due to its recognized ability to form amalgams with Au. The inclusion of Cd in this work is a very positive and promising aspect for future multielemental detection.

This work has no major suggestions. My congratulations to the authors, excellent work.

I only recommend just a few very minor things.

Please include more data regarding the calibration curve. It is understood that the calibration curve is external mode and that it was not necessary standard addition or matrix match methods, but it is not clear how this conclusion was reached. Were the slopes compared with those types of curves? Regarding the linear range, this is provided in concentrations, but it is important to know the injection volume or, provide the linear range in mass.

Please include the operating currents of the Hg and Cd HCL lamps.

Please include information about RSD determination. What is the concentration of the standard or sample used to determine the precision?

Author Response

Dear Reviewer:

Thank you so much for your valuable comments from the reviewer. Herein, we have made modification carefully in line with your advice using the “Track Changes” function in the revised manuscript. In the meantime, we have put together the revisions and explanations as follows:

Question 1: Please include more data regarding the calibration curve. It is understood that the calibration curve is external mode and that it was not necessary standard addition or matrix match methods, but it is not clear how this conclusion was reached. Were the slopes compared with those types of curves? Regarding the linear range, this is provided in concentrations, but it is important to know the injection volume or, provide the linear range in mass.

Answer: Thanks for your comment. The regression equations of Hg and Cd calibration curves have been added in the manuscript (Line 254). In Figure 6, the Hg and Cd signal peaks were compared using the same concentration of the standard solution and real rice sample. And the results showed the peak heights between the rice sample and standard solution were approximately consistent (1:0.96 for Hg and 1:0.99 for Cd), which proved free of interference on Hg and Cd analysis. Thus, the standard calibration strategy was performed. I am sorry that this part was not explained clearly in the manuscript, so the relevant content has been added. Since Figure 6 had demonstrated that there is no significant difference in signal peak height between the solid sample and the standard solution, we did not additionally try standard addition or matrix match methods. And in the revised manuscript, all sample introduce volume or mass were replenished, and I am very sorry for my carelessness. Please check it.

Question 2: Please include the operating currents of the Hg and Cd HCL lamps.

Answer: Thanks for your warm reminding. The operating currents of Hg and Cd HCL lamps are 3 mA and 5 mA, respectively, and they have been added in the revised manuscript. Please check it.

Question 3: Please include information about RSD determination. What is the concentration of the standard or sample used to determine the precision?

Answer: Thanks for your suggestion. To calculate the RSDs of Hg and Cd in standard solutions and grain samples, the 10 μL Hg and Cd mix solution (500 μg/L) and 10 mg R-3 rice sample (Hg: 44 μg kg-1, Cd: 870 μg kg-1) were selected, respectively. And the concentration of the standard or sample used to determine RSD has been added in the revised manuscript. (Line 258-259)

Reviewer 2 Report

A portable and rapid Hg-Cd analyzer was proposed by Authors for grain analysis using a metal-ceramic heater electrothermal vaporizer, an on-line catalytic pyrolysis furnace, a composite Pt/Ni trap, and a home-made miniature atomic absorption spectrometry. The method achieved limits of detection of 1.1 μg/kg and 0.3 μg/kg for Hg and Cd, respectively, using 20 mg grain samples without the need for sample digestion or chemical usage. The newly designed folded light path using CCD allowed for simultaneous Hg and Cd analysis while eliminating grain matrix interference through the utilization of a catalytic pyrolysis furnace with aluminum oxide fillers coupled with a composite Pt/Ni trap.

I would recommend intensive English grammar and style corrections.

There are a few minor errors in the text that can be corrected for clarity:

"metal-ceramic heater (MCH) electrothermal vaporizer (ETV)" should be "metal-ceramic heater (MCH)-based electrothermal vaporization (ETV)"

"To enhance the sensitivity, a folded light path was newly designed for AAS using CCD for simultaneous Hg and Cd analysis;" could be rephrased as "To enhance sensitivity, a new folded light path was designed for simultaneous Hg and Cd analysis using CCD in AAS."

Also

All references should be corrected to follow journal template, and especially should be described as follows (see Instruction for Authors), depending on the type of work:

e.g. Journal Articles:

1. Author 1, A.B.; Author 2, C.D. Title of the article. Abbreviated Journal Name Year, Volume, page range.

Author Response

Dear Reviewer:

Thank you so much for your valuable comments from the reviewer. Herein, we have made modification carefully in line with your advice using the “Track Changes” function in the revised manuscript. In the meantime, we have put together the revisions and explanations as follows:

Question 1: "metal-ceramic heater (MCH) electrothermal vaporizer (ETV)" should be "metal-ceramic heater (MCH)-based electrothermal vaporization (ETV)"

Answer: Thanks for your warm reminding. The “metal-ceramic heater (MCH) electrothermal vaporizer (ETV)” has been changed to metal-ceramic heater (MCH)-based electrothermal vaporization (ETV) according to your requirement, and please check the revised manuscript.

Question 2: "To enhance the sensitivity, a folded light path was newly designed for AAS using CCD for simultaneous Hg and Cd analysis;" could be rephrased as "To enhance sensitivity, a new folded light path was designed for simultaneous Hg and Cd analysis using CCD in AAS."

Answer: Thanks for your warm reminding. The sentence has been modified in the revised manuscript.

Question 3: All references should be corrected to follow journal template, and especially should be described as follows (see Instruction for Authors), depending on the type of work:

  1. Author 1, A.B.; Author 2, C.D. Title of the article. Abbreviated Journal Name Year, Volume, page range.

Answer: Thanks for your reminding. All references have been checked and modified in the revised manuscript. Please check it.

Reviewer 3 Report

Dear Authors,

Your manuscript entitled "Rapid and portable detection of Hg and Cd in grain samples based on novel catalytic pyrolysis composite trap coupled with miniature atomic absorption spectrometry" is quite interesting and novel, it is scientific sound and could be interesting for readers of journal.

However, I have noticed some serious analytical flaws, hence, below find some queries that need to be addressed point by point.

1. Lines 123-127: Please provide Hg uncertainties for Hg in all CRMs. 2. Lines 123-127 Hg values are below LOQ of the method. Authors say that LOD is equal to 1.1 μg/Kg. This means that LOQ is for about 3.3 μg/kg so they cant accurately determined it. Please explain. 3. In figure Hg absorbance is in 0.005 to 0.030 values whereas in Figure 4 optimization Hg absorbance is 0.1 to 0.3 values. Please explain the significant differences. 4. Authors declare that have recoveries for Hg 97-106% (Abstract section). In Rice(GBW(E)100360) the recovery is 3.6*100/3=120%. Please correct, revise accordingly.. 5. In Table 3 there other LOD values for this work and are worst than those provided in the abstract section. Furthermore RSD values are not explained how they were calculated.
References are appropriate, minor spelling and grammar mistakes exist.
Major revision is required.

Author Response

Dear Reviewer:

Thank you so much for your valuable comments from the reviewer. Herein, we have made modification carefully in line with your advice using the “Track Changes” function in the revised manuscript. In the meantime, we have put together the revisions and explanations as follows:

Question 1: Lines 123-127: Please provide Hg uncertainties for Hg in all CRMs.

Answer: Thanks for your warm reminding. I am very sorry that the Hg uncertainties for Hg were not provided in the recognition certificates of GSB-24 and GSB-3 CRMs, and only a Hg reference values were provided. Therefore, to gain the Hg concentrations, the GSB-24 and GSB-3 CRMs were detected by the direct Hg analyzer (Model 5E-HGT2321), the Hg values and uncertainties were also shown in Table 2.

Question 2: Lines 123-127 Hg values are below LOQ of the method. Authors say that LOD is equal to 1.1 μg/Kg. This means that LOQ is for about 3.3 μg/kg so they cant accurately determined it. Please explain.

Answer: Thanks for your comment. The Hg LOQ of this method was 3.6 μg/kg (Line 257), it is indeed higher than the reference values of CRMs of GSB-24 and GSB-3, which can’t be accurately determined by this proposed method. But the spiked results of CRMs are consistent with that by ICP-MS, and results of other CRMs and real samples were also adequate in agreement with the certified values and that by ICP-MS, indicating accuracy of this method. Otherwise, the results which are below the method LOQ were marked with NQ in Table 2. And the grain samples with above 3.6 μg kg-1 Hg can be measured accurately. This proposed LOQ is significantly lower than the maximum allowable concentration in grain (0.02 mg kg-1) in Chinese standard (China National Standard: GB 2762-2022), and can be applicable to the real grain sample analysis for Hg and Cd monitoring.

Question 3: In figure Hg absorbance is in 0.005 to 0.030 values whereas in Figure 4 optimization Hg absorbance is 0.1 to 0.3 values. Please explain the significant differences.

Answer: Thanks for your comment. In Figure 3 and Figure 6, the Hg absorbance is approximately 0.0157 for 10 mg R-3 rice sample (44 μg kg-1 Hg), but in Figure 4, the Hg relative absorbances are normalized with the signals at 30 W set as 1 for 500 μg L-1 Hg solution (10 μL). Thus, the absorbances shown differences. It is our negligence that is not clearly marked in the Figures, and the Figures have been modified.

Question 4: Authors declare that have recoveries for Hg 97-106% (Abstract section). In Rice(GBW(E)100360) the recovery is 3.6*100/3=120%. Please correct, revise accordingly.

Answer: Thanks for your valuable reminding. In this work, the recoveries are the ratio of Hg and Cd levels measured by the proposed method after spiking to the added values of Hg and Cd. I am very sorry that I didn’t in detail describe the calculation method in the manuscript, and this calculation method has been added in the manuscript. Please check it.

Question 5: In Table 3 there other LOD values for this work and are worst than those provided in the abstract section. Furthermore RSD values are not explained how they were calculated.

Answer: Thanks for your valuable reminding. We are very sorry for our careless mistake, the LOD values for this work in Table 3 been corrected to Hg: 1.1 μg/kg and Cd 0.3 μg/kg. And the RSDs were calculated by repeating 5 measurements in grain samples. The calculate method was shown in line 258, and please check it.

Question 6: References are appropriate, minor spelling and grammar mistakes exist.

Answer: Thanks for your warm suggestion. English language has already been revised for good comprehension. Please check the revised manuscript

Round 2

Reviewer 3 Report

Authors replied to my comments adequately and applied all my suggestions. I am satisfied with their response and improvements in their article, hence, I suggest publication in its current form.